# Emergency Communications Based on Throughput-Aware D2D Multicasting in 5G Public Safety Networks

**DOI:** 10.3390/s20071901

**Published:** 2020-03-29

**Authors:** Mengjun Yin, Wenjing Li, Lei Feng, Peng Yu, Xuesong Qiu

**Affiliations:** State Key Laboratory of Networking and Switching Technology, Beijing University of Posts and Telecommunications, Beijing 100876, China; yinmengjun@bupt.edu.cn (M.Y.); wjli@bupt.edu.cn (W.L.); fenglei@bupt.edu.cn (L.F.); xsqiu@bupt.edu.cn (X.Q.)

**Keywords:** public safety network, device-to-device, Hungarian algorithm

## Abstract

Emergency communications need to meet the developing demand of equipment and the complex scenarios of network in public safety networks (PSNs). Heterogeneous Cloud Radio Access Network (H-CRAN), an important technology of the 5th generation wireless systems (5G), plays an important role in PSN. H-CRAN has the features of resource sharing and centralized allocation which can make up for resource shortage in emergency communications. Therefore, an emergency communications strategy based on Device-to-device (D2D) multicast is proposed to make PSN more flexible and rapid. Nearby users can communicate directly without a base station through D2D. This strategy may guarantee high speed data transmission and stable continuous real-time communications. It is divided into three steps. Firstly, according to the distance between users, the alternative cluster head is divided. Secondly, two kinds of cluster head user selection schemes are developed. One is based on terminal power and the other is based on the number of extended users. Last but not least, the Hungarian Algorithm based on throughput-aware is used to channel multiplexing. The numerical results show that the proposed scheme can effectively extend the coverage of PSN and optimize the utilization of resources.

## 1. Introduction

Emergency communications, necessary communications methods to ensure emergency rescue and necessary communications, are carried out normally after disaster. It can also be used in the case of natural or man-made emergencies, such as the surge in communication demand due to important holidays [1]. This network for emergency communications can be called a public safety network (PSN). With the development of mobile communications equipment and the complex changes of emergencies, this information transmitted in PSN is not only voice services, but also stable real-time videos. In addition, the communications equipment carried by the rescue team is difficult to replenish in time in some emergency situations. Thus, continuous communications has become one of the factors that limits the progress of rescue [2]. The Heterogeneous Cloud Radio Access Network (H-CRAN) is proposed as a framework of 5th Generation mobile communications technology (5G). In addition, it is considered as one of the core technology solutions to solve the problem of 5G [3]. In H-CRAN, High-Power Nodes (HPNs) are compatible with traditional cellular networks and take charge of supporting coverage. Meanwhile, the Radio Remote Unit (RRU) is deployed in specific areas to enhance local network capacity and rate [4]. Furthermore, all resources are centralized by the central management unit for the unified control. In emergency communications, H-CRAN has capabilities of resource-sharing and dynamic on-demand. These configurations enable limited network resources to be used more efficiently and flexibly. Therefore, it has been the critical research study to make emergency communications have the following properties: extensive coverage, high reliability, real time, and terminal energy savings.

There are many ways to improve communications quality and expand communications coverage in emergency scenarios such as increasing transmitting power of the base station (BS) [5], adjusting antenna tilt of the BS [6], using Coordinated Multiple Point (CoMP) to enhance received power [7], and so on. Although these methods can increase coverage area, they will also increase energy consumption. They occupy more resources, have less flexibility, and have limited compensation capabilities. The relay is usually used to expand BS coverage area. For example, an air base station (ABS) can be deployed in large post-disaster scenarios. These methods can greatly improve the network coverage and communications quality but result in long deployment time and high network cost. At present, these methods cannot meet the increasing demand of emergency communications.

Due to the evolution of 5G technology, D2D communications have been recognized widely. This technology has a lot of potential, such as improving system performance, enhancing user experience, and expanding mobile Internet applications [8]. D2D communications are direct communications between wireless terminals based on cellular networks [9]. Therefore, we hope to propose emergency communications based on throughput-aware D2D multicasting. Firstly, mobile terminals are clustered. Then, a cluster head is selected from each cluster. The cluster head will be the relay of BS. Finally, the remaining terminals in cluster are connected with the relay by D2D [10]. This PSN structure can reduce the terminal energy consumption and guarantee the quality of service in the whole communications process. It has the following three advantages: (1) The destroyed communications network can be restored rapidly through D2D multicasting to ensure the progress of a rescue mission going on smoothly. (2) In the scenario when a communications blind area occurs due to terrain or building block. The terminal located in the blind area can communicate with other terminals through one hop or multi-hop D2D multicasting communications. (3) The transmitting power of the wireless terminal can be reduced by using D2D short-distance communications to replace long-distance communications from a BS.

At present, there are still some problems in existing research. First of all, they were proposed for one BS scenario without considering the interference of multi-cells [11,12,13]. Second, in the actual scenario, the electricity consumption of cluster head is bigger than other users [14]. However, the clustering method and the selection of the cluster head lacks consideration of the terminal power, which will affect the smoothness progress of rescue mission in practical applications. For example, Yaacoub [15,16] uses the traversal method to cluster terminals. However, the computational complexity is high, and they did not consider the impact of the power consumption when selecting the cluster head. In addition, PSN resources are limited due to equipment damage. A scheme of resource allocation is necessary. Reference [17] summarizes and discusses various energy efficiency and resource allocation methods, but it is not applied to PSN. Reference [18] proposed a cluster-based resource allocation mechanism. This mechanism uses a rate-based allocation algorithm for sub-channel allocation and a distributed power allocation algorithm for power allocation. It can achieve a considerable effect in an instant. However, it cannot adapt to the user’s rapid movement in PSN. 

In order to improve the above problems and realize emergency communications in the public safety network, this paper proposes emergency communications based on throughput-aware D2D multicasting in 5G public safety networks. This method consists of three parts: the alternative cluster head set partitioning, the cluster head selection, and the resource allocation. First, based on the location of the terminal and its maximum service radius, all the terminals can be divided into multiple clusters. Each terminal in the cluster can provide services to others. In addition, it is also within the maximum service range for others. Secondly, two cluster head selection methods are proposed: maximize remaining battery electricity (MinRE) and maximize user number (MaxUN), and a resource allocation method based on a Hungarian algorithm put forward for PSN. In the simulation, the effectiveness is better than the D2D clustering method in the paper [11]. It is superior at extending the duration and energy-saving. 

The innovation of this paper is as follows:

(1) In this paper, an emergency communications mechanism based on D2D multicast technology is proposed. This mechanism can increase the communications range of PSN, reduce the network resource consumption, improve the service quality of users, and apply flexibly to various practical environments. 

(2) A clustering method and two cluster head selection mechanisms are designed to optimize the allocation of PSN resources. It reduces the system’s energy consumption and realize the continuous communications. 

(3) A channel allocation method suitable for emergency communications scenarios is proposed to optimize the allocation of network resources.

The rest of paper is organized as follows: The Section 2 describes emergency communications based on throughput-aware D2D multicasting, which includes the application scenario and specific detailed process. In the Section 3, system models and problem models are presented. The Section 4 is a distance-based D2D multicast formation scheme and the resource allocation method based on a Hungarian algorithm. Simulation environment and results are described in the Section 5. In addition, the conclusion is provided in the Section 6.

## 2. D2D Emergency Communications Mechanism

### 2.1. Application of D2D in PSN

Cell outage compensation is an important solution to maintain network communications in the event of a disaster. It is also a key technology for PSN [19]. D2D technology, as a common communications enhancement technology, has been widely studied in the field of emergency communications [20,21,22]. In order to improve D2D emergency communications, a complete compensation process needs to be established. 

D2D communications is a technology that allows the terminals to communicate directly through multiplexing the cell resources. As shown in Figure 1, the H-CRAN architecture is composed of HPN, RRU, Base Band Unit (BBU), and Central Management Unit (CMU). All resource is regulated uniformly by CMU in the BBU pool. The CMU can allocate resources directly without considering the details of the connection. D2D technology can implement the following types of communications in 5G PSN: resource reuse in a single HPN, inter-regional direct communications between HPNs, self-organizing communications without HPN, and extended communications between HPN coverage and the outside [23]. The advantages of D2D are as follows: high speed and low latency, efficient resource utilization, ease congestion and reduce network transmission load, coverage extension, and so on. Under the characteristics of resource sharing and centralized allocation of H-CRAN, it is hoped that the extended coverage function of D2D can compensate the users with outages. Finally, emergency communications based on D2D multicast are realized.

Emergency command center of the PSN receives information from different rescue teams. The rescue equipment includes equipment vehicle, ambulance, and fire engine. These equipment and ordinary users are referred to as user equipment (UE). After filtering and analyzing, the data are sent to the rescue teams. For example, the pictures and videos shot by rescue helicopters can be distributed to rescue teams to determine the situation in a disaster area. The D2D multicasting can reduce the network load, reduce the path loss of edge users, and expand the transmission range. As shown in Figure 1, when a HPN or RRU interrupts, the adjacent cells service the outage users by D2D multicast communications. In addition, if the power of cluster head is low, the continuous communications can be achieved by converting cluster heads. PSN terminals include rescue vehicles, rescue equipment, and general users, which will be collectively recorded as UEs in the following text.

Generally, when D2D multicasting transmits by downlink, it will strongly interfere the neighboring cell’s UEs. When it uses uplink, the neighboring users will cause a strong interference to the receiver. This paper considers the application of D2D extended coverage in the PSN. The neighboring users are in outage. Therefore, the D2D multicasting uses the downlink to achieve the purpose of compensation. Then, UEs can be divided into four categories:T-DU (Transmit-UE): T-DU is a cluster head of a D2D cluster. As the transmitting end of D2D multicasting, it is responsible for forwarding the HPN/RRU signals to the receiving users.AT-DU (Alternative Transmit-UE): AT-DU can be used as the T-DU for D2D multicasting. After selecting the T-DU, the rest of the AT-DU receives the information as the receiving users.R-DU (Receive-UE): As a receiving user, R-DU receive signals transmitted by T-DU.CU (Cellular UE): The users directly covered by the HPN/RRU, which transmit uplink and downlink data through the HPN/RRU.

The PSN channel resources are limited. Multiplexing the uplink channel of the CU can reduce the resource pressure. In order to ensure the fairness of channel resource allocation, we assume that each D2D cluster can only reuse one channel and one channel can only be multiplexed by only one D2D cluster. As shown in Figure 2, the T-DU1 multiplexes the data of R-DU1 and R-DU2 with the channel #1 resource of CU1. In addition, the T-DU2 multiplexes the uplink channel #2 resource of CU2. In addition, The T-DU3 is multiplexed with the uplink channel #3 resource of CU3. Due to the multiplexing of uplink resources, the T-DU interferes the HPN/RRU. At the same time, CU interferes the R-DUs.

### 2.2. The Process of D2D Emergency Communications

The process of emergency communications based on throughput-aware D2D multicasting in 5G PSN is shown in Figure 3. The sustainable D2D emergency communications mechanism is divided into four steps: information collection, calculation of AT-DU, T-DU selection, and resource allocation. First, UE information is collected, including all UEs in the coverage area of the HPN/RRU and adjacent outage area. The filtered UEs who are close to the adjacent HPN/RRU can expand the coverage. So the 30% edge UEs are selected to be clustered according to the geographical location. Then, AT-DU is selected according to a cluster head selection scheme as far as possible to ensure that there are enough AT-DUs to be selected to maintain communications. Next, the cluster head selection mechanism is used to select the appropriate user in the AT-DUs as the T-DU. Among the edge users without AT-DU, the T-DU should also be found. Finally, CU channel multiplexing is selected by the Hungarian algorithm. As the communications continue, the UE data are changing, including T-DU power and UE location information. When the current T-DU power is insufficient, the next AT-DU is searched to serve as T-DU. 

## 3. System Model and Problem Model

### 3.1. Variable Declaration

The channel model, SINR, data rate, and problem models are expounded in this section. Before describing these models, the variables are identified in Table 1.

The received power of UE is calculated by the transmit power *P_t_* and the channel gain *H*. If a UE can only provide services through HPN, its received power is calculated through the HPN link. The D2D link is preferentially used when a UE can also be served by T-DU to reduce the burden of PSN. 

### 3.2. Channel Model

#### 3.2.1. HPN to UE

The channel model of the HPN to UE used in this paper is the modified Cost231-Hata model, as shown in Equation (1). $f_{c}$ is the carrier frequency in the unit of MHz. $h_{\mathrm{te}}$ is the effective height for HPN antenna, and $h_{\mathrm{re}}$ is the height for UE. In addition, $d_{H_{k},U_{j}}$ stands for the distance between *H_k_* and *U_j_*. $a(h_{\mathrm{re}})$ is a correction factor for UE height, $a(h_{\mathrm{re}})=(1.11\mathrm{lg}f_{c}-0.7)h_{re}-(1.56\mathrm{lg}f_{c}-0.8)$. In addition, $C_{M}$ is a correction factor for metropolitan centers. Those values are shown as Table 2. Therefore, $H_{H_{k},U_{j}}^{\mathrm{HPN}}$ is calculated by Equation (2):(1)$$L^{HPN}\left(d_{H_{k},U_{j}}^{}\right)=46.3+33.9\mathrm{lg}f_{c}-13.82\mathrm{lg}h_{\mathrm{te}}-a(h_{\mathrm{re}})+(44.9-6.55\mathrm{lg}h_{\mathrm{te}})\mathrm{lg}d_{H_{k},U_{j}}+C_{M},$$
(2)$$H_{H_{k},U_{j}}^{\mathrm{HPN}}=(L^{\mathrm{HPN}}(d_{H_{k},U_{j}}))+L_{\mathrm{fading}},$$

#### 3.2.2. UE to UE

Assume that all users are in an outdoor environment in PSN. Thus, the path loss mode for D2D is Outdoor to Outdoor. The D2D communications path loss model between the transmitting end U*_i_* and the receiving end U*_j_* may be represented by Formulas (3)–(6) [24]. The value of *L*_urban_ depends on the environment as Table 2. $d_{U_{i},U_{j}}$ is the distance between *U_i_* and U*_j_* with the unit of m. $f_{\mathrm{UE}}$ is 2 GHz. The shadow fading *L*_fading_ in Line of Sight (LoS) and Non Line of Sight (NLoS) is 7 dB. Therefore, $H_{U_{i},U_{j}}^{D2D}$ is calculated by (7):(3)$$L^{D2D}(d_{U_{i},U_{j}})=\{\begin{array}{ccc} L_{\mathrm{LoS}}^{D2D}(d_{U_{i},U_{j}}) & , & d_{U_{i},U_{j}}\leq 44.2\\ L_{\mathrm{NLoS}}^{D2D}(d_{U_{i},U_{j}}) & , & d_{U_{i},U_{j}}\geq 64.2\\ L_{\mathrm{trans}}^{D2D}(d_{U_{i},U_{j}}) & , & 44.2<d_{U_{i},U_{j}}<64.2 \end{array},$$
(4)$$L_{\mathrm{LoS}}^{D2D}(d_{U_{i},U_{j}})=32.45\mathrm{dB}+20\log_{10}(f_{\mathrm{UE}})+20\log_{10}(d_{U_{i},U_{j}}/1000),$$
(5)$$L_{\mathrm{NLoS}}^{D2D}(d_{U_{i},U_{j}})=9.5\mathrm{dB}+45\log_{10}(f_{\mathrm{UE}})+40\log_{10}(d_{U_{i},U_{j}}/1000)+L_{\mathrm{urban}},$$
(6)$$L_{\mathrm{trans}}^{D2D}(d_{U_{i},U_{j}})=L_{\mathrm{LoS}}^{D2D}(44.2)+\left(L_{\mathrm{NLoS}}^{D2D}(64.2)-L_{\mathrm{LoS}}^{D2D}(44.2))\cdot (d_{U_{i},U_{j}}-44.2\right)/20,$$
(7)$$H_{U_{i},U_{j}}^{D2D}=L^{D2D}(d_{U_{i},U_{j}})+L_{\mathrm{fading}},$$

### 3.3. SINR and Data Rate

According to the specifications of 3GPP [25,26], the downlink transmission herein adopts Orthogonal Frequency Division Multiplexing (OFDM). Assume that orthogonal spectrum resources are used by neighboring users in a single cell. Therefore, the interference between UEs who do not participate in multiplexing is negligible. $\sigma^{2}$ is Gaussian white noise. The SINR *γ*_u_ is counted as Label (8) for the downlink of CU and T-DU. For a CU*_j_*, whose uplink is multiplexed by T-DU*_k_*, the SINR is counted by (9). T-DU*_k_* transmits data signals to *NI ^k^*
$\mathrm{R-D}U_{i}^{k}$. T-DU*_k_* uses the channel of CU*_j_* so that CU*_j_*’s uplink communications will interfere with R-DUs in the cluster. Thus, the SINR of $\mathrm{R-D}U_{i}^{k}$ is counted as (10):(8)$$\gamma_{U_{j}}=\frac{P_{\mathrm{HPN}}^{t}H_{H_{k},U_{j}}^{HPN}}{\sigma^{2}},$$
(9)$$\gamma_{CU_{j}}^{D}=\frac{P_{CU_{j}}^{t}H_{H_{k},CU_{j}}^{HPN}}{P_{T-DU_{k}}^{t}H_{T-DU_{k},H_{k}}^{D2D}+\sigma^{2}},$$
(10)$$\gamma_{CU_{j},R-DU_{i}^{k}}^{D}=\frac{P_{T-DU_{k}}^{t}H_{T-DU_{k},R-DU_{i}^{k}}^{D2D}}{P_{CU_{j}}^{t}H_{CU_{j},R-DU_{i}^{k}}^{\mathrm{HPN}}+\sigma^{2}}.$$

Due to the nature of multicast, the communications quality of the T-DU*_k_* is determined by the worst channel quality among *NI^k^* receiving devices. Therefore, according to the fragrance theorem, the data rate can be obtained separately. Formula (11) is for the downlink of CU and T-DU, (12) is for CU*_j_*, and (13) is for the *k*^th^ D2D cluster. Whether a user can be served by a link is determined by the value of the data rate. If at least one link’s data rate is greater than the threshold value γ_th_, it means that this user can access the network. The user is called an active user:(11)$$R_{U_{j}}=B\log_{2}(1+\gamma_{U_{j}}),$$
(12)$$R_{CU_{j}}^{D}=B\log_{2}(1+\gamma_{CU_{j}}^{D}),$$
(13)$${R^{D}}_{CU_{j},T-DU_{k}}=B\cdot NI^{k}\cdot \log_{2}(1+\underset{i=1\cdots NI^{k}}{\min}{\gamma^{D}}_{CU_{j},R-DU_{i}^{k}}).$$

### 3.4. Problem Model

#### 3.4.1. Emergency Communications Based on Throughput-Aware D2D Multicasting

Emergency communications based on throughput-aware D2D multicasting in 5G public safety networks can be split into two questions: a D2D multicasting formation problem and Resource allocation problem. The multicasting formation scheme is used to generate D2D AT-DU sets and cluster head selection. In an emergency scenario, the division of the AT-DU set is based on the user’s location information. The specific algorithm is described in Section 4. 

#### 3.4.2. Cluster Head Selection Problem

There are two ways to select T-DU. The first one is maximize the remaining battery electricity scheme (MaxRE). It selects the T-DU according to the terminal power, slowing down the power consumption of PSN. The other one is to maximize the extended UE number scheme (MaxUN). It selects the T-DU on the number of expandable UEs.
MaxRE: There are two ways to select T-DU. The first is to maximize the remaining battery electricity scheme (MaxRE). It selects the T-DU according to the terminal power, slowing down the power consumption of PSN. The other one is to maximize the extended UE number scheme (MaxUN). It selects the T-DU on the number of expandable UEs:
(14)$$\begin{array}{ll} & T-\mathrm{DU}(Ca^{n})=\arg\;\underset{Ca^{n}}{\max}RE({\mathrm{AT}-\mathrm{DU}}_{i})\\ s.t.\; & \\ & \mathrm{AT}\_{\mathrm{DU}}_{i}\in Ca^{n} \end{array}$$MaxUN: The goal of this method is to extend active UEs in outage. A step function ε(x) is introduced when counting the number of users, where x≥0, ε(x)= 1; else, ε(x)= 0. If *U_j_* is not within a HPN’s coverage but in AT-DU*_i_*, *U_j_* is an extended UE of AT-DU*_i_.* All users are counted to get the value of UN(AT-DU*i*). In addition, the AT-DU with the largest number of extended users is selected as the current T-DU:(15)$$\begin{array}{ll} & T-\mathrm{DU}\left(Ca^{n}\right)=\arg\;\underset{Ca^{n}}{\max}\;UN\left({\mathrm{AT}-\mathrm{DU}}_{i}\right)\\ s.t. & \\ & UN\left({\mathrm{AT}-\mathrm{DU}}_{i}\right)=\sum_{U_{j}\in U}\varepsilon (d(U_{i},U_{j})<d_{\max}^{U})\times \varepsilon (P_{H_{k}}^{t}H_{H_{k},U_{i}}^{\mathrm{HPN}}<P_{th})\\ & U_{i}\in Ca^{n},U_{j}\in U \end{array}$$

#### 3.4.3. Resource Allocation Problem

There are *NK* T-DUs. Each T-DU serves *NI ^k^* R-DUs. In addition, *NC* CUs are directly served by HPN. We define a matrix of *NK***NC*, $\Delta ={\left[\delta_{k,j}\right]}_{NK*NC}$ to indicate channel resource allocation. $\delta_{k,j}$ is a binary variable. If the value of $\delta_{k,j}$ is 1, it indicates that the T-DU in the *k*^th^ D2D cluster multiplexes the uplink resource of the CU*_j_*. Otherwise, the value of $\delta_{k,j}$ is 0.

In order to ensure the quality of service for both CUs and D2D users, the maximum capacity is set to the goal of resource allocation. It is assumed that the T-DU uses the maximum transmit power in the initial state. It guarantees the largest number of users accessing the network. When the T-DU transmit power is determined, the downlinks do not interfere with each other. The remaining system throughput is composed of the uplink throughput of all CU and the downlink throughput of D2D users. It depends on both the CU throughput and the R-DU throughput (16), wherein $\Delta R_{U_{j}}^{D}=R_{CU_{j}}^{D}+{R^{D}}_{CU_{j},T-DU_{k}}-R_{U_{j}}^{}$:(16)$$\begin{array}{cc} R(P_{CU}^{t},\delta_{kj}) & =\sum_{j=1}^{NC}\sum_{k=1}^{NK}\delta_{kj}(R_{CU_{j}}^{D}+{R^{D}}_{CU_{j},T-DU_{k}})+\sum_{j=1}^{NC}(1-\sum_{k=1}^{NK}\delta_{kj})R_{U_{j}}\\ & =\sum_{j=1}^{NC}\sum_{k=1}^{NK}\{\delta_{kj}(R_{CU_{j}}^{D}+{R^{D}}_{CU_{j},T-DU_{k}}-R_{U_{j}})\}+\sum_{j=1}^{NC}R_{U_{j}}\\ & =\sum_{j=1}^{NC}\sum_{k=1}^{NK}\delta_{kj}\Delta R_{U_{j}}^{D}+\sum_{j=1}^{NC}R_{U_{j}} \end{array},$$

The value of the system throughput indicator $R(P_{CU}^{t},\delta_{kj})$ is related to CU transmit power $P_{CU}^{t}$ and channel allocation matrix $\Delta ={\left[\delta_{k,j}\right]}_{NK*NC}$. Therefore, the objective function can be defined as maximizing channel multiplexing increments (17):(17)$$\begin{array}{l} \arg\max\sum_{j=1}^{NC}\sum_{k=1}^{NK}\delta_{kj}\Delta R_{U_{j}}^{D}\\ s.t.\\ \;\sum_{k=1}^{NK}\delta_{kj}\leq 1\;,\sum_{j=1}^{ND}\delta_{kj}\leq 1\;\\ \;\gamma_{CU_{j}}^{D}\geq \gamma_{th},\;\;{\gamma^{D}}_{CU_{j},T-DU_{i}^{k}}\geq \gamma_{th}\\ \;P_{CU_{j}}^{t}\leq P_{\max}^{t}\\ \;i\in \left\{NI\right\},j\in \left\{NC\right\},k\in \left\{ND\right\} \end{array}$$
wherein (1) the resource of each CU can only be multiplexed by at most one D2D cluster. A D2D multicast group can only reuse one CU resource. (2) Both D2D users and CUs must meet the quality of service requirements. (3) The uplink transmit power of the CU shall not exceed the upper limit. Then, $\Delta R_{U_{j}}^{D}$ is only affected by $P_{CU}^{t}$. That is, each $\Delta R_{U_{j}}^{D}$ can calculate a maximum value $\max\Delta R_{U_{j}}^{D}(\delta_{kj})$ through the above constraints. When the channel selection is completed, the final objective function value of the system can be obtained.

## 4. Algorithm Design and Application

### 4.1. Distance-Based AT-DU Clustering Method

Based on the distance between UEs, an AT-DU clustering method is proposed. This method only considers the location information between the UEs. This simplifies the information exchange between the UEs in the clustering process. In this clustering mode, each terminal in the AT-DU set can serve as a T-DU. The method utilizes the location information of the UEs and the maximum service radius $d_{\max}^{U}$ to determine the cluster. When the distance between *U_1_* and *U_2_* is *d*(*U_1_, U_2_*)< $d_{\max}^{U}$, *U_1_* and *U_2_* can be AT-DUs. However, *U_3_* is needed to meet *d*(*U_1_, U_3_*)< $d_{\max}^{U}$ and *d*(*U_2_, U_3_*)< $d_{\max}^{U}$. Therefore, each user within the set can replace others as AT-DU. The specific algorithm flow is as follows (Algorithm 1):**Step1**: The information of all users is processed. In addition, 30% edge users are added to the search domain eU.**Step2**: When eU is not empty, the farthest node Ua is selected. Add all UEs in the maximum service radius of node Ua to the temporary collection Temp. The UEs in Temp are arranged in ascending order according to the distance to Ua. When Temp is empty, Ua can be a finished cluster Cu. If Temp is not empty, go to **Step3**.**Step3**: Initialize Ua’s cluster Cu={Ua}. Traverse the elements of Temp in order. Ui is an arbitrary element in Cu. If d(Ui , Ub)< $d_{\max}^{U}$, Ub is added into Cu, and Cu=Cu∪Ub.**Step4**: The current Cu is an AT-DU set and stored into Ca. After that, all the nodes in Cu are moved out of the search domain eU back to the Step2 loop until the search field eU is empty. Finally, output the total AT-DU set Ca.


**Algorithm 1: Distance-based AT-DU clustering method**

**Input**
 U:The location information of UEs
 HPN: The location information of HPNs
 eU: The set of edge UEs (Search domain)
**Output**
 Cu: Current alternative cluster head set
 Ca: AT-DU set including some Cu sets**Step1** 1 Calculate eU←{*n_i_*|*i*=1…n}**Step2** 2 While eU≠∅ do3  Ua=find max(d(eU,HPN) )4  Temp ← find(d(*U_a_,U_b_*)<$d_{\max}^{U}$)5  Sort(Temp), Cu={*U_a_*}6  If Temp=∅, go to **Step4**7  If Temp≠∅, go to **Step3****Step3** 8  for i=1…| Temp |9   *U_b_* = Temp(i)10   If *d*(*U_i_*, *U_b_*)<$d_{\max}^{U}$, Cu=Cu∪*U_b_*11  end for**Step4** 12  eU=eU-Cu13 End while

### 4.2. Application of the Hungarian Algorithm

#### 4.2.1. Structure Bipartite Graph

A bipartite graph G=(V,E,W) is used to model the matching problem between D2D user clusters and CU channels. V is a vertex set. It can be divided into two sets of D2D cluster set {D} and the cellular user set {C}. E is a collection of edges, and the vertices associated with each edge belong to {D} and {C}, respectively. W represents the weight value $\max\Delta R_{U_{j}}^{D}(\delta_{kj})$ of each edge. When the value of |D| is large, it is not convenient to allocate channels. Therefore, the Hungarian algorithm is proposed. The Hungarian algorithm needs to satisfy three conditions to solve the problem: 1) The objective function is finding the minimum value; 2) The efficiency matrix is a square matrix; 3) All elements in the matrix are greater than 0. Multiplexing rate increment $\max\Delta R_{U_{j}}^{D}(\delta_{kj})$ is not necessarily to be a square matrix. Therefore, the original bipartite graph needs to be transformed.

#### 4.2.2. Bipartite Transformation

A bipartite graph is constructed as shown in Figure 4. Each cellular user cluster CU has channel resources R. If the *j*^th^ D2D cluster can reuse the *i*^th^ CU resource, there is an edge with the weight of the multiplexing rate increment $\max\Delta R_{U_{j}}^{D}(\delta_{kj})$ between them. When the bipartite graph is asymmetrical, it is necessary to construct virtual vertices and corresponding virtual edges. The weight of the virtual edges is the maximum of the weight of all edges. In an emergency scenario, the CU number *NC* is greater than D2D clusters number *NK*. Then, there are two cases when the bipartite graph is transformed:NC = α*NK, α∈Z+. The number of CU is a positive integer multiple of the D2D clusters. At this point, it is necessary to expand the vertices number NK to α*NK. This means that (α-1)*NK virtual cluster vertices are added as shown in Figure 4a.NC = α*NK+β, α, β∈N+. The number of CU is not a positive integer multiple of the D2D clusters. It is needed to expand the cluster vertices number NK to (α+1)*NK. This means that α*NK virtual cluster vertices are added. The number of {C} is extended to NC+NK-β. This means that NK-β virtual CU vertices are added as Figure 4b.

#### 4.2.3. Hungarian Algorithm

After performing the bipartite transformation, the weight matrix becomes a square matrix. Then, the Hungarian algorithm can be used to solve the maximum weight matching problem of the bipartite graph [27,28]. The algorithm specific process is as follows (Algorithm 2):**Step1:** An efficiency matrix $H={\left[h_{ij}\right]}_{\alpha *NK\times \alpha *NK}$ or $H={\left[h_{ij}\right]}_{(\alpha +1)*NK\times (\alpha +1)*NK}$ is generated based on the expanded bipartite graph. In order to facilitate the calculation, NT records the number of both rows and columns of the matrix. Then, $H={\left\lfloor h_{ij}\right\rfloor }_{NT\times NT}$. Matrix update times t = 0, Ht = H.**Step2:**$r_{i}=\underset{\forall j}{\min}h_{ij}^{t}$ is found as the minimum value of each row of the matrix Ht. In addition, we use $h_{ij}^{\prime }=h_{ij}^{t}-r_{i}$ to get a new matrix $H^{\prime }={\left\lfloor h_{ij}^{\prime }\right\rfloor }_{NT\times NT}$. Then, $c_{j}=\underset{\forall i}{\min}h_{ij}^{\prime }$ is found as the minimum value of each column of the matrix H’. In addition, a new matrix $H^{\prime \prime }={\left[h_{ij}^{\prime \prime }\right]}_{NT\times NT}$ is created by $h_{ij}^{\prime \prime }=h_{ij}^{\prime }-c_{j}$. Matrix update times t = t + 1, Ht = H’’.**Step3:** NL is the number of the least horizontal or vertical lines that is used to cover all $h_{ij}^{t}$ = 0 in Ht. If NL < NT, go to step4. Otherwise, go to **Step5**.**Step4:** Find the minimum value $\Psi =\min h_{ij}^{t}$ which is not covered by horizontal and vertical lines in $h_{ij}^{t}$. Subtract the minimum value Ψ from each line that is not covered to get a new matrix H’. In addition, the minimum value Ψ is added to each column covered to obtain a new matrix H’’. Matrix update times t = t + 1, Ht = H’’. Then, return to **Step3**.**Step5**: The matrix *H^t^* obtained at this time is the final change result. It is needed to choose the optimal allocation result from *H^t^*. The *h_ij_* = 0 is marked with a row or column with the least 0 in the matrix *H^t^*. In addition, all 0 elements of this row or column are removed. It continues to mark the remaining elements until *NT* zeros are marked. At this point, the marked position is recorded as 1, and the remaining positions are recorded as 0. The result is the best matching matrix M. Finally, the virtual channel in M and the node connected to it are deleted to obtain the optimal channel allocation matrix Δ.


**Algorithm 2: Channel assignment based on Hungarian algorithm**

**Input**
 C: cellular user set
 D: D2D cluster set
 H: Efficiency matrix
**Output**
 M: Best matching matrix (0-1)
 Δ: Channel allocation matrix (0-1)**Step1** 1  t=0, H(T, i, j)= h_ij_;**Step2** 2 for i=1, α*NK3  r_min=min(H(i,:)); H_1(i,j)= h_ij_ -r_min; 4 end for5 for j=1, α*NK6  c_min=min(H_1(:, j)); H_2(i, j)= H_1(i, j)- c_min;7 end for8 T=T+1, H(T, i, j) =H_2(i, j);9 NL=0, r_line=0, c_line=0**Step3** 10 for i=1, α*NK11  r_zero=find (H(i,:)==0);12 end for13 for j=1, α*NK 14  c_ zero=find (H(:,j)==0);15 end for16 Subgraph G(r_G, c_G) cover all 0 elements17 NL= r_G+c_G;18 If NL<NT go to **Step4**19 If NL=NT go to **Step5****Step4** 20 Subgraph K=H-G, Min_value= min(K(i, :))21 for i=1, α*NK22  H_1(i,:)= H (i, :) - Min_value;23 end for24 for j=1, α*NK25  H_2(:,j)= H_1(:, j)+Min_value;26 end for27 T=T+1, H(T, i, j) =H_2(i, j);**Step5** 28 M=zeros(NT,NT), Number of marks Nm=0;29 While Nm<NT do30  Find the row or column with the least 0, H(a, b)=0;31  Mark M(a, b)=1; 32 end while33 Get M, Delete the virtual node to get Δ.

## 5. Simulation and Analysis

### 5.1. Parameter Settings

MATLAB is used as a simulation experiment platform to verify the performance. Simulation and analysis include parameter settings, the impact on user status, and communications duration, and the resource reuse effects comparison of different algorithms. The user status mainly considers the number of active users, D2D clusters, and the D2D users. The comparison of resource reuse includes no-reuse scheme, random scheme [29], heuristic algorithm [30], and the Hungarian algorithm proposed in this paper. The key parameters of the simulation are shown in Table 3. The channel model of the HPN and the UE use the Dense-urban/High-rise scene. The original area contains nine evenly distributed HPN nodes. Five HPNs were unexpectedly interrupted in an emergency scenario. This paper hopes to realize the user’s fast communications connection through D2D multicast emergency communications by the remaining four active HPNs. In this area, 100–1000 user nodes are randomly deployed to compare the application effects of different user densities. 

### 5.2. Compensation Effect

The communications network link status is restored quickly through D2D multicast communications. On one hand, the network collects user information of power and location in disasters. The control center calculates of the AT-DU, cluster head selection (MaxRE/MaxUN), and recalculates the overlay user. In order to evaluate the recovery effects, the user status is compared including the number of active users before and after emergency measures, the number of D2D clusters, and the number of users at the D2D receiving end. On the other hand, the large power consumption of the user relay limits the sustainable transmission time. Therefore, it is necessary to compare the user access durations of different cluster head selection algorithms. 

First, the comparison of user states under different system users is shown in Figure 5. The abscissa is the total number of system users that is gradually increased from 100 to 1000. The following conclusions can be seen from the figure:In terms of the overall trend, the D2D emergency measures have greatly improved the user active status in the network. There is no obvious difference between the MaxRE and MaxUN cluster head selection methods.As the number of users increases, the total number of active users changes significantly. The larger the number of system users, the greater the proportion of active users increased through D2D multicast communications. The proportion of UE in outage increases as the user density increases. More interrupted users can access the network through D2D.The growth rate of the T-DU is gradually decreasing. As the user density increases, the number of AT-DU increases. However, the T-DU will only choose one of them as the transmitter of D2D. The T-DU without AT-DU reduces. As a result, the proportion of the T-DU in the total users gradually increased from 10% (100 total users) to 19.75% (800 total users) and stabilized at around 20%. The change of active users is mainly due to the increase of R-DU.The total number of active users of MaxUN is slightly larger than that of MaxRE. The number of D2D receivers of the two schemes is similar. In addition, MaxUN has less T-DU. Thus, the increase of MaxUN’s total active users comes from the users who have not been covered.

Then, the number of users is fixed at 800. The effect of the T-DU maximum transmit power on the overall user state after clustering is compared with Figure 6. The maximum transmit power of the T-DU is gradually increased from 0 mW to 200 mW. Take MaxRE as an example: when the maximum transmit power increases, the coverage of T-DU is larger. The number of active users is also rising. Therefore, the total number of active users has gradually increased from 94.6% (0 mW) to 99.6% (160 mW). The increased transmit power led to a significant increase in R-UD numbers. The number of D2D clusters has remained stable. Since the choice of the T-DU is only related to the geographic location of the users, its slight float is the result of expanding coverage by R-DU which acts as T-DU. The total active users of MaxRE and MaxUN are almost the same. The number of T-DU and R-DU of MaxUN is slightly greater than that of MaxRE. This is consistent with the target set by the cluster head selection method. 

The maximum transmit power of the UE is set to 40 mW. The number of users of D2D continuous communications is observed with time as shown in Figure 7. The abscissa is the analog communications time unit T. The power consumption of the T-DU is large. In addition, its power is gradually reduced to the critical point. The dotted line in the figure indicates the D2D transmission performed by MaxRE and MaxUN without AT-DU. The solid line indicates the schemes with the AT-DU. At T = 60, the T-DU starts to turn into the R-DU due to insufficient remaining power. As a result, the number of users accessing the network begins to decline. During T = 180–240, the solid lines stabilize again due to the role of the AT-DU. At T = 240, the number of active users of all schemes decrease continuously. Only a small number of users with multiple AT-DU maintain continuous transmission. Until T = 380, none of the users have enough power to be a T-DU. In addition, the number of active users returns to the original state. The setting of the AT-DU greatly prolongs the time for users to communicate continuously. In addition, it guarantees the communications situation of most users before the disaster reconstruction.

### 5.3. Resource Reuse Algorithm Comparison

Due to the limited channel resources in an emergency scenario, the Hungarian algorithm is used to multiplex the uplink channel of the CU based on throughput-aware. Multiplexing effect is compared with the full random, heuristic algorithm as shown in Figure 8. The abscissa is the total number of system users that is gradually increased from 100 to 1000. The maximum transmit power of T-DU is 80 mW. The blue line named ‘Before disaster’ stands for the situation under normal circumstances with enough resources. It is also the maximum limit of throughput. The dotted line named ‘Disaster, No reuse’ stands for the situation without reuse resources. Assume that the current system resources can only satisfy 80% of the users of the system. 

Values of throughput and average data rate per UE in the system are compared in Figure 8a,b. Figure 8c,d compare these values of R-DUs. It can be seen that the gap is gradually increased between the throughput of D2D multicast transmission with resource reuse and that before the disaster. When the number of users is 800, the throughput is nearly 25 Mbps, which is less than before outage. Compared with the no reused D2D compensation strategy, the advantage of reuse also increases gradually. This is on the account of allowing more users to successfully connect to the network by resource reuse. Different algorithms and cluster head selection schemes are compared after amplification. It can be seen that the throughput of MaxRE is 1.3% higher than MaxUN. The results of Hungarian algorithm are better than a heuristic algorithm, and the heuristic algorithm is better than the random one. The best one of “MaxRE, Hungarian” is 1.9% higher than that of the “MaxUN, Random”. In addition, the average data rate per UE decreases gradually as the number of users increases. When the number of users was 800, it was down 12.2% from 100 users. This is due to the interference caused by a reuse channel. When the total number of users increases, the proportion of users reused also increases. The number of users with interference increases, and the average data rate decreases. It can be seen that the throughput of the MaxRE is higher than MaxUN. The results of the Hungarian algorithm are better than a heuristic algorithm, and the heuristic algorithm is better than the random one.

Figure 8c,d are compared R-DU throughput and average data rate per R-DU of R-DUs. It can be seen that the gap of throughput of R-DU between now and pre-outage increased gradually. When the number of users is 800, the throughput is nearly 25 Mbps less than pre-outage. Combining with the analysis of the total throughput of the system, it can be seen that the change of the total throughput is mainly affected by the R-DU’s. D2D multicast enables UE to be reconnected to the network after interruption. However, its data rate is lower than the rate of direct transmission by HPN/RRU. Similarly, different algorithms and cluster head selection schemes are compared. MaxRE is higher than MaxUN. The Hungarian algorithm is better than heuristic algorithm, and the heuristic algorithm is better than the random one. The data rate per R-DU shows an upward trend. As the UE density increases, more UEs can be connected to D2D links with high data rates. However, algorithm comparison of data rate per R-DU yields different results from the R-DU throughput. As the total number of users accessing the network increases through the Hungarian algorithm, the mean value of R-DU decreases. In general, the data rate per R-DU of MaxRE is higher than that of MaxUN. In addition, the results of the Hungarian algorithm are better than other algorithms. The validity of the proposed algorithm is further verified.

### 5.4. Results Summary

The D2D emergency solution proposed in this paper can quickly restore the communications status of users. Depending on user distribution and limit of transmit power of T-DU, the active UE rate can be 94.6–99.6%. AT-DUs are used to extend the duration of continuous communications which provide more adequate preparation time for subsequent emergency measures. Comparing with no AT-DU, this scheme can make communications effect double. System throughput acts as the optimization target. It can improve the communications quality of the whole UE. The system’s interference is reduced. The PSN is closest to the situation before the disaster. The scheme reduces the resource pressure in an emergency scenario by multiplexing the uplink channel resources of the CU. The Hungarian algorithm has great advantages over other algorithms both in terms of throughput of the system and data rate per UE communication.

In addition, MaxRE and MaxUN, the two cluster-head selection schemes provided in this paper, are suitable for different scenarios. MaxRE is fit for PSN with a high demand of network stability. Since high-power T-DU is selected, the MaxRE can maintain communications for a longer time without switching cluster heads frequently. For example, in the scenario of an emergency rescue team requiring a more stable network, it is better for MaxRE to be applied to this situation. MaxUN is selected for PSN with high active user rate requirements. It can satisfy the active UE of the largest number. MaxUN can be used together with other emergency measures such as air base station emergency deployment. It takes a certain amount of time for an air base station to be deployed. This MaxUN D2D multicast strategy can be used to achieve short-term emergency communications before deployment of air base stations.

## 6. Conclusions

Based on throughput-aware D2D multicasting, a PSN emergency communications scheme in H-CRAN was analyzed. This emergency solution could expand the coverage of network quickly and effectively. It ensured the number of active users and the service quality of PSN. When a T-DU’s power was too low to service, an AT-DU from candidate sets could be quickly selected to ensure continuous communications. Finally, the D2D groups used the uplink channel of CU to implement resource multiplexing. The Hungarian algorithm was used for channel allocation. Therefore, it was significant for the PSN to apply throughput-aware D2D multicasting in communications H-CRAN.

With the development of technology, the network has become more diversified. According to complex network structure, we hope to put forward a resource optimization algorithm for a better performance. In addition, the T-DU selection method designed only considers the terminal power or extends the UE number. If these factors can be considered together, a better effect may be achieved. Of course, using new technologies such as air base stations and millimeter wave to improve emergency efficiency is also a future research direction in 5G.

## Figures and Tables

**Figure 1 sensors-20-01901-f001:**
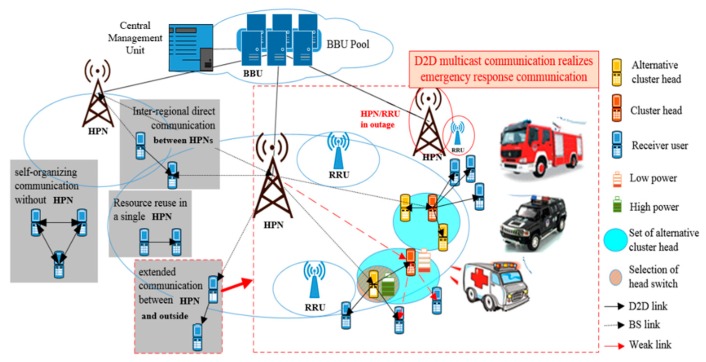
Device-to-device (D2D) technology and the application in PSN.

**Figure 2 sensors-20-01901-f002:**
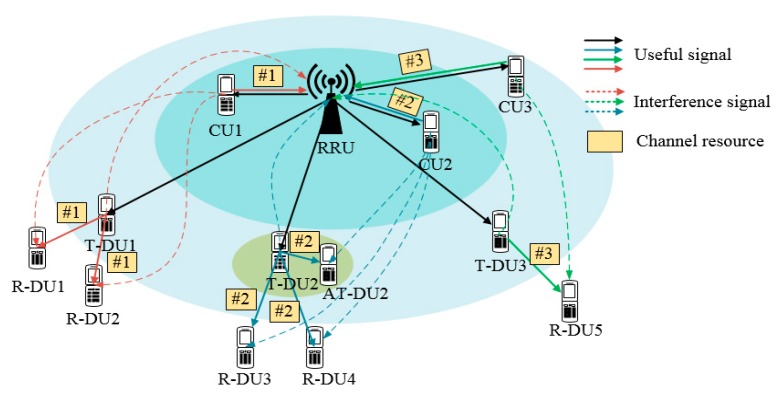
UE classification and channel multiplexing.

**Figure 3 sensors-20-01901-f003:**
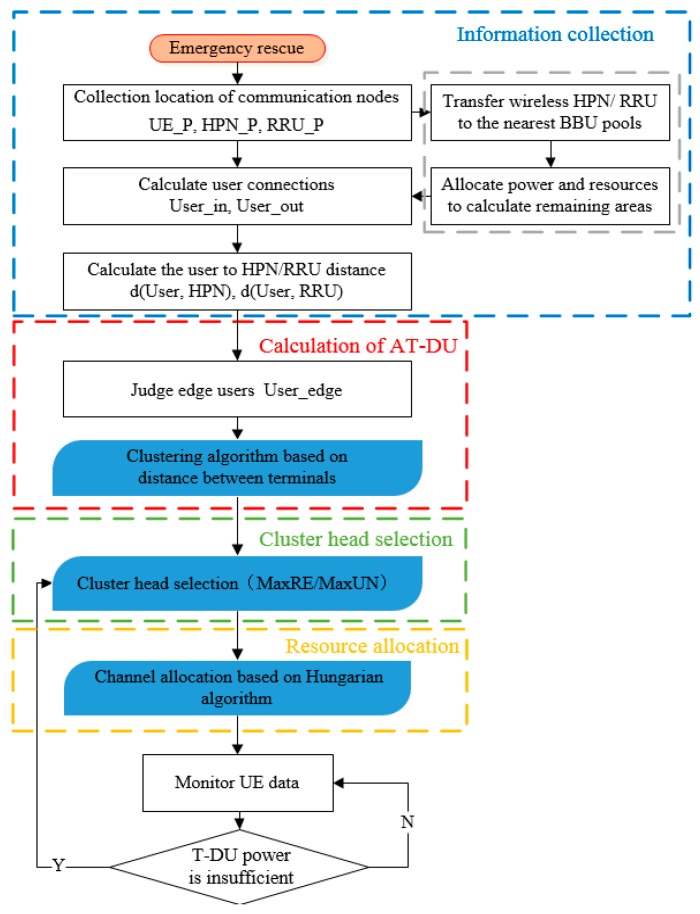
The process of D2D multicasting emergency communications.

**Figure 4 sensors-20-01901-f004:**
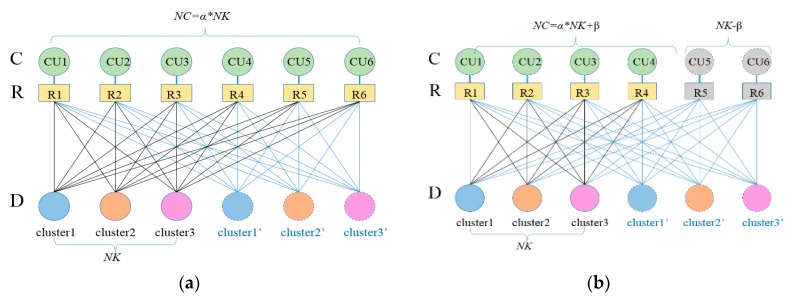
Bipartite transformation of the original matrix: (**a**) NC=α*NK; (**b**) NC=α*NK+β.

**Figure 5 sensors-20-01901-f005:**
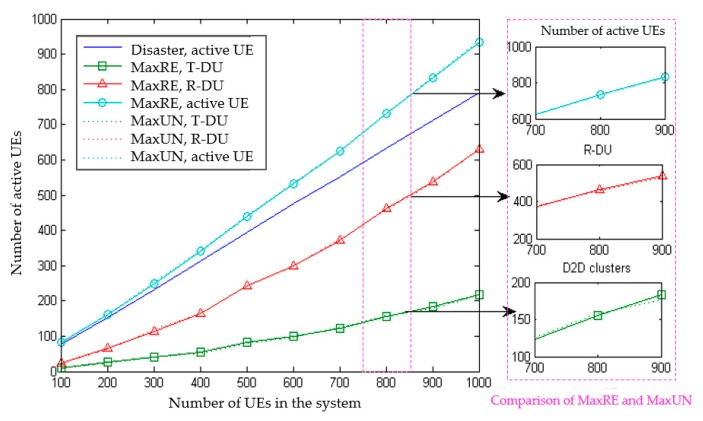
The comparison of UE states under different system user numbers.

**Figure 6 sensors-20-01901-f006:**
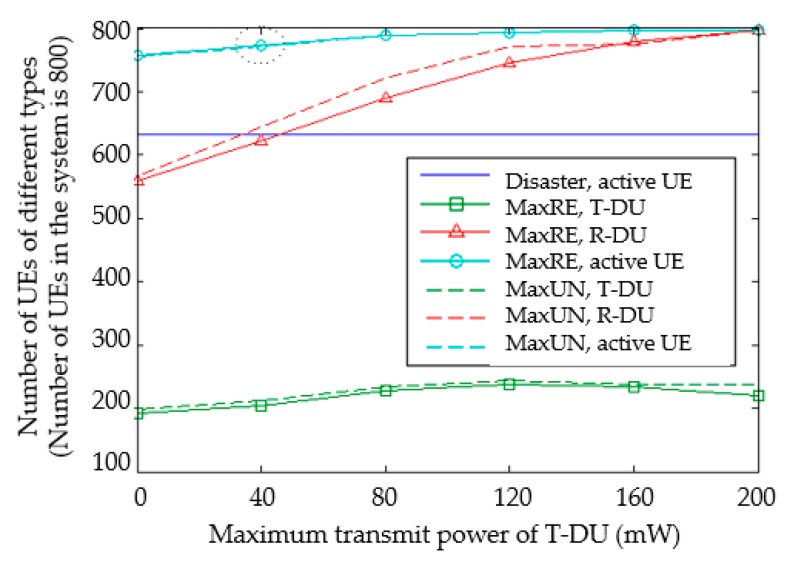
The comparison of UE states under different transmit power.

**Figure 7 sensors-20-01901-f007:**
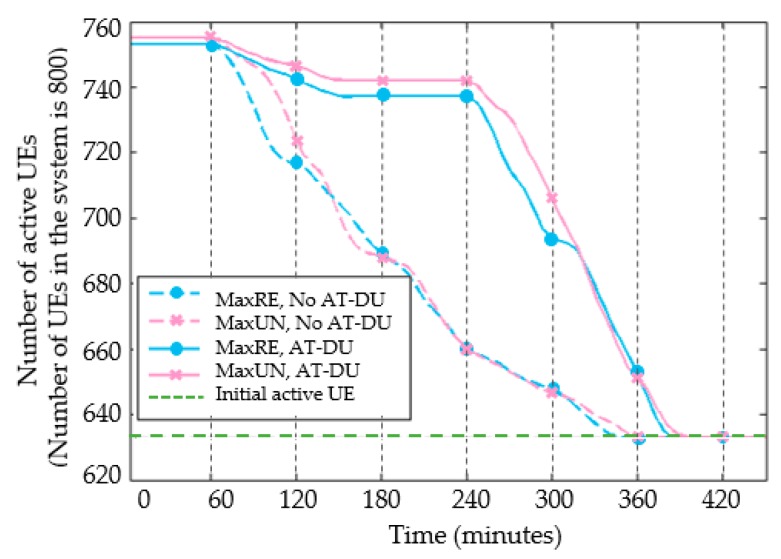
Comparison of active users over time.

**Figure 8 sensors-20-01901-f008:**
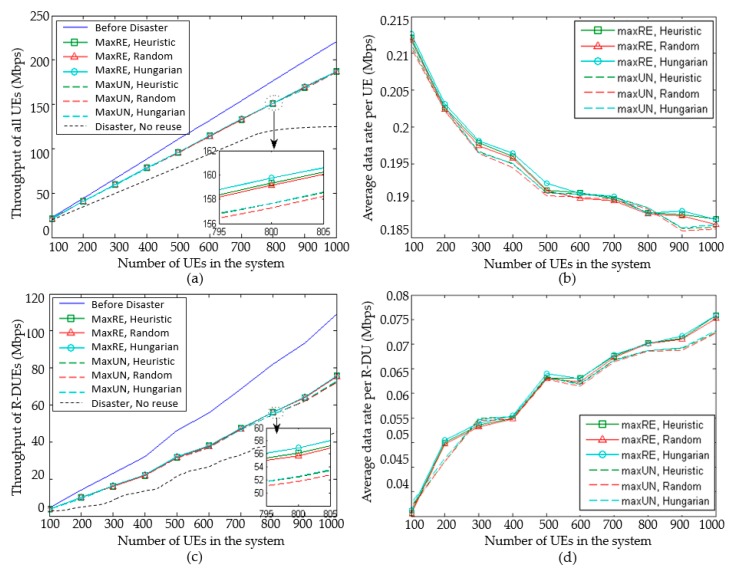
The impact of resource allocation methods: (**a**) throughput of all users; (**b**) average data rate per UE; (**c**) throughput of R-DUs; (**d**) average data rate per R-DU.

**Table 1 sensors-20-01901-t001:** Variable declaration.

Variable	Notation
*H_k_*	The No. *k* High-Power Node (HPN).
*U_i_*	The No. *i* user equipment (UE).
$P_{x}^{t}$	The transmit power of node *x*.
$P_{x,y}^{r}$	The received power of node *y* from *x*.
*d_x,y_*	The distance between *x* and *y*.
$d_{\max}^{U}$	Maximum distance for Device-to-device (D2D).
$H_{H_{k},U_{j}}^{\mathrm{HPN}}$	The channel gain from *H_k_* to *U_j_*.
$H_{U_{i},U_{j}}^{D2D}$	The channel gain from *U_i_* to *U_j_*.
*Cu^n^*	The No. *n* cluster head partitioning set.

**Table 2 sensors-20-01901-t002:** Parameters’ value of *C*_M_ and *L*_urban._

Environment	Suburban	Urban	Dense Urban or High-Rise
CM	−12.28 dB	0 dB	3 dB
Lurban	0 dB	6.8 dB	2.3 dB

**Table 3 sensors-20-01901-t003:** The key parameters of simulation.

Parameter	Value
The transmit power of HPN $P_{H_{k}}^{t}$	46 dBm
The number of HPN *K^0^*	9
The number of active HPN *NK*	4
The number of simulated users *NI*	100–1000
Carrier frequency $f_{c}$	2.6 GHz
HPN antenna effective height $h_{\mathrm{te}}$	45 m
UE antenna effective height $h_{\mathrm{re}}$	1.5 m
*C* _M_	3 dB
*L* _urban_	2.3 dB
*L* _fading_	7 dB
Receive power threshold of UE *P_th_*	−105 dBm
The transmit power of UE $P_{U_{i}}^{t}$	0–200 mw
$d_{\max}^{U}$	102 m–180 m
SINR threshold of UE and HPN *γ*_th_	6 dB
White Gaussian Noise $\sigma^{2}$	−154 dBm
Power ratio for communications	50%
UE battery capacity	2000 mAh
Basic UE power consumption	80 mW
Minimum communications power of UE	200 mAh

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
