# Peer review of "Emergency Communications Based on Throughput-Aware D2D Multicasting in 5G Public Safety Networks"

_sensors, 2020, doi:10.3390/s20071901_

Round 1

Reviewer 1 Report

  • The article focuses on the communication aspect of using an alleged 5G network to implement a public rescue system. It uses 2.6 GHz. I could not see the benefit of using this 5G since the frequency is close to the one used by ISM WiFi. Why couldn't regular 4G network be used? If sticking to 5G, why not go for millimeter-wave frequencies whose bandwidth to video transmission, as suggested by the authors, is one of the focus for emergency protocols?
  • "we hope to realize the emergency" - misused verb, instead of realize I guess authors meant to implement, propose, etc.
  • "the power of the cluster head cost faster than others" - didn't understand the sentence, what is the cost doing here?
  • "References [12]" - is it one of several of them? Plural or singular?
  • H-C-RAN not defined in the text.
  • BBU not defined in the text.
  • HPN not defined in the text.... Please make sure all these similar words are defined the reader cannot guess/google them all.

Reviewer 2 Report

The paper presents a three-step scheme for Emergency Communication Based on Throughput-Aware D2D Multicasting in 5G Public Safety Networks

The mathematical apparatus as well as the comparativ performance evaluation show some promising results, however the paper has several shortcomings:

  • The English should absolutely be revised. Especially the abstract is incomprehensible and does not entice a reader to go through the paper
  • Authors fail to define a number of acronyms. Nowhere throughout the paper are acronyms H-C-RAN and HPN explained and they hold a significant amount of space throughout the text
  • The analysis of existing research is superficial and the provided references are insufficient for strengthening the argument of the authors as regards to the challenges they intend to solve in the paper. 
  • The link between the cluster selection and the resource allocation algorithms is not clear. 
  • The authors mention in the introduction they propose MinEC and MaxUN cluster head selection methods, but in section 3, they propose MaxRE and MaxUN methods (!?). Please clarify
  • The relevance of Fig 5 is not clear. The two axis labels are especially confusing  (UE number vs System UE number ?). The authors give in the text some more explanations, but it is still not clear.
  • Figure 6 shows a number of values. It is not clear which T-DU is taken into consideration. The chart shows ~200 D2D clusters, so there are ~200 T-DUs.
  • Figure 7 does not show any measurement unit for time. Please clarify.
  • It is not clear why authors evaluate their resource allocation algorithm solely on throughput. In any case, the analysis should be more thorough, especially since there are different values of throughput for different devices. It would be worthwhile to add a section discussing these results. In particular it is not clear on what proof the authors make their claim that "the emergency solution proposed (...) can repair the disaster area without manual intervention". This is based on what? Throughput values ? 

Other minor flaws:

  • It is not clear how one can "multiplex a single channel"
  • Conclusion section should be in past tense

Recommendations:

  • Please redo the figures and add clarity to the axis labels. In particular in Fig 5 "System UE number" should read "Number of UEs in the system". Also "UE number" should be perhaps "Number of Access UEs". Same for each figure 5-8.
  • Clarify what you mean by "Access UEs" and why that is relevant
  • Make sections 1 and 2 more coherent by clearly stating the challenges and showing how your paper solves them.
  • Define the acronyms the first time they appear in the text
  • Enhance section 5.3 and add some more details (how many D2D clusters have been considered, etc.)
  • Please add subsection 5.4 discussing overall the sections 5.2 and 5.3. Refer to results in both sections and analyze the results for different scenarios and provide recommendations for future implementation in a real network. This would highly improve the quality of the paper.

Reviewer 3 Report

In this paper, the authors are proposing a D2D based scheme for providing emergence services in publica safety networks. The paper’s technical contribution is acceptable however the paper lacks appropriate presentation and organization structure.  The paper’s presentation style should be improved since no careful presentation of the notations, equations are given. There are many incomplete and grammatically wrong sentences. I advice the authors to proofread the paper with a native speaker. Therefore I recommend major revision.

  • Add indentations to algorithms to make them readable on page 11 and 9.
  • Sentence at the end of abstract is incomplete
  • The authors should not start sentence with “And ” in most of the text
  • Who proposed on page 2 line 62? Which references?
  • The abbreviations are not clear. For example what is H-C-RAN on page 3 line 109, HPN on page 4 line 136, CU on page 4 line 144?
  • The sizes of notations are not same and in some cases not readable, e.g. on page 6 line 185, on page 8 line 237, etc. I advice authors to use appropriate text editor.
  • The references are not appropriate for journal’s standards. In some cases it is given as footnote (e.g. on page 6 line 190) and in some cases as normal.
  • The equations are not well aligned. For example on page 8 Eq. (15)
  • On page 10 line 280 Fig. 4 is referenced but it is not given.
  • Is Fig. 1 on page 10 Fig.4?
  • What is chapter on page 11 line 324
  • Are there any references for the algorithms defined on page 11 line 326? More details about those algorithms that are compared are needed.
  • The figures legend marking are not distinctive in simulation results section.
  • No numerical comparisons are given in Section 5.3 when different algorithms are compared.
  • Increase the fonts of figures, e.g. Fig. 8, etc
  • Try to adjust the y axis as log scale as the different between algorithms are not clear in Fig. 8

Minor comments:

  • On page 2 line 84, “It is superior in ”
  • On page 2 line 47 “This technology”
  • On page 2 line 91, “It reduces”
  • On page 2 line 68, 70 “Reference [11]” and “Reference [12]”
  • On page 4 line 162, “to maintain communication”, line 159 “who are close to”
  • On page 8 line 231 “the maximum capacity”
  • On page 8 line 240, “objective function can be defined”
  • On page 13 line 374 sentence is incomplete
  • On page 13 line 364, “who has not been covered”
  • In titlte of Section 5.3 “Reuse Algorithm”

Round 2

Reviewer 1 Report

Article has been significantly improved in the present version.

Author Response

Thank reviewers for the advices. The language of this article has been improved.

Reviewer 2 Report

The authors have mostly addressed the previous comments.

However the English should still be reviewed. E.g. in the first phrase of the abstract :  The Emergency communications need to meet the developing demand of equipment.

I believe the correct wording is communications (a system for transmitting information, technology) not communication (as in human communication) 

Same goes for the paper title actually. It should read "Emergency communications based on ..."

Reviewer 3 Report

The authors have responded to my previous comments.

Author Response

(The authors gave the same response as above.)
